# Assessment of the Safety and Probiotic Properties of *Enterococcus faecium* B13 Isolated from Fermented Chili

**DOI:** 10.3390/microorganisms12050994

**Published:** 2024-05-15

**Authors:** Jingmin Xiao, Cai Chen, Zhuxian Fu, Shumin Wang, Fan Luo

**Affiliations:** 1Institute of Qinghai-Tibetan Plateau, Southwest Minzu University, Chengdu 610041, China; jingminxiao@outlook.com (J.X.); 13975959752@163.com (C.C.); f2835757820@163.com (Z.F.); 2College of Animal and Veterinary Sciences, Southwest Minzu University, Chengdu 610041, China; wang1310973830@outlook.com

**Keywords:** probiotic properties, safety, *Enterococcus faecium*, whole-genome sequences, gut flora, metagenomics

## Abstract

*Enterococcus faecium* B13, selected from fermentation chili, has been proven to promote animal growth by previous studies, but it belongs to opportunistic pathogens, so a comprehensive evaluation of its probiotic properties and safety is necessary. In this study, the probiotic properties and safety of B13 were evaluated at the genetic and phenotype levels in vitro and then confirmed in vivo. The genome of B13 contains one chromosome and two plasmids. The average nucleotide identity indicated that B13 was most closely related to the fermentation-plant-derived strain. The strain does not carry the major virulence genes of the clinical *E. faecium* strains but contains *aac(6′)-Ii*, *ant (6)-Ia*, *msrC* genes. The strain had a higher tolerance to acid at pH 3.0, 4.0, and 0.3% bile salt and a 32.83% free radical DPPH clearance rate. It can adhere to Caco-2 cells and reduce the adhesion of *E. coli* to Caco-2 cells. The safety assessment revealed that the strain showed no hemolysis and did not exhibit gelatinase, ornithine decarboxylase, lysine decarboxylase, or tryptophanase activity. It was sensitive to twelve antibiotics but was resistant to erythromycin, rifampicin, tetracycline, doxycycline, and minocycline. Experiments in vivo have shown that B13 can be located in the ileum and colon and has no adverse effects on experiment animals. After 28 days of feeding, B13 did not remarkable change the α-diversity of the gut flora or increase the virulence genes. Our study demonstrated that *E. faecium* B13 may be used as a probiotic candidate.

## 1. Introduction

*Enterococcus* was widely present in traditional fermented foods, such as milk [1], meat [2], cheese [3], fermented sausage [4], fermented vegetables [5,6], soy milk [7], and other foods [6,8]. Because of its strong proteolytic and esterolytic properties, strong tolerance, and yielding acid, *Enterococcus* is even intentionally added to the process of food fermentation [9,10]. For instance, *Enterococcus* is frequently used in the preparation of traditional cheeses in Mediterranean countries, including Greece, Italy, Spain, and Portugal, to accelerate food ripeness and aroma development [11,12]. *Enterococcus faecium*, belonging to lactic acid bacteria, is a relatively unique type of probiotic. They are not only widely used in fermented foods but also extensively used in animal health [13], such as restoration of microbiota balance of the gastrointestinal tract (GIT) with antibiotic-induced dysbiosis [14,15,16], antiviral activity [17], antitumor effect [18,19], cholesterol-lowering effect [20], and immune regulation [21,22,23].

However, unlike other probiotics, such as *Lactobacillus* and *Bifidobacterium*, *E. faecium* possesses strong antibiotic resistance, and its resistance is still evolving. Recent research showed that the clinical infection caused by *E*. *faecium* has continued to rise. They often show resistance to antimicrobials, including β-lactams, high levels of aminoglycosides, and glycopeptides [24]. Even antimicrobial resistance against vancomycin has been reported [25,26]. Many studies have shown that the resistance of *E. faecalis* is closely related to its survival environment. *E. faecalis*, from the clinic, generally exhibited strong resistance [27,28], while strains from fermented foods had weaker resistance [4]. But antimicrobial-resistant *E. faecium* with low pathogenicity still affects these immunocompromised patients and possibly causes systemic infections because it limits the choice of effective antibiotics. So even low-pathogenic *E. faecium* from food sources may be unsafe. Therefore, the safety of using *E. faecalis* as a probiotic in the fields of food and healthcare has been questioned by people [29,30]. It is essential to evaluate its probiotic properties, especially its safety, before use.

The *E. faecium* B13 in this study was screened from chili and has been proven to produce bacteriocins and promote animal growth in previous studies [31]. To determine whether the strain has application possibilities in food and animal healthcare, the safety and probiotic properties of *E. faecium* B13 must be examined based on these guidelines related to probiotics, including those issued by the United States Food and Drug Administration (FDA) and the World Gastroenterology Organization (WGO).

## 2. Materials and Methods

### 2.1. Strains and Culture Conditions

*E. faecium* B13 used in this study was previously isolated from fermented pickled chili (preserved at Southwest Minzu University, Chengdu, China). It was cultured still at 37 °C in DeMan-Rogosa-Sharpe (MRS) medium, containing (per liter) 10 g tryptone, 5 g yeast extract, 2 g diammonium hydrogen citrate, 20 g glucose, 5 g anhydrous sodium acetate, 2 g K_2_HPO_4_·3H_2_O, 0.58 g MgSO_4_·7H_2_O, 0.25 g MnSO_4_·H_2_O, and 10 g beef extract. *Escherichia coli* 8099 and *Staphylococcus aureus* ATCC 25923 purchased from Chengdu Pengshida Experimental Supplies Co., Ltd. (Chengdu, China) were used as indicator strains and cultured at 37 °C for 24 h in Luria–Bertani (LB) medium, containing (per liter) 10 g tryptone, 5 g yeast extract, and 10 g NaCl.

### 2.2. Whole-Genome Sequencing

#### 2.2.1. Genomic DNA Extraction

Genomic DNA was extracted from the culture broth (24  h) using the Bacterial DNA Kit (Omega Biotek, Norcross, GA, USA) and quantified by TBS-380 Picogreen (Invitrogen, Carlsbad, CA, USA) according to the manufacturer’s protocol.

#### 2.2.2. Illumina and PacBio Sequencing

In this study, the *E. faecium* B13 strain genome was sequenced by a combination of PacBio RS II and Illumina NovaSeq 6000, respectively.

For Illumina pair-end sequencing of this strain, purified genomic DNA was sheared into smaller fragments with 300~500 bp by Covaris M220 (Covaris, Woburn, MA, USA), and genomic libraries of *E. faecium* B13 were constructed using the TruSeq™ Nano DNA Sample Prep Kit (Illumina, San Diego, CA, USA). The whole genome sequencing was performed on an Illumina NovaSeq 6000 (150 bp*2, Shanghai BIOZERON Co., Ltd., Shanghai, China) using the Truseq SBS Kit (Illumina, California, USA) with 300 cycles.

Moreover, genomic DNA was processed into 15–20 kb fragments by the G-tubes method and sequenced using a PacBio BS(Sequel) II instrument(PacBio, Menlo Park, CA, USA) following the Pacbio standard protocol. The data were assembled using unicycler version 0.4.8. The protein sequences were predicted by GeneMarkS (version 4.17), and the COG database was used to annotate the functions of the predicted open reading frames.

#### 2.2.3. Average Nucleotide Identity

42 strains of *E. faecium* from different sources in the NCBI database were selected and their average base similarity was compared with *E. faecium* B13 using the ANI Calculator online analyzer (https://www.ezbiocloud.net/tools/ani) (accessed on 6 October 2023).

#### 2.2.4. Identification of Virulence Factors and Antibiotic Resistance Genes

The virulence factor database (VFDB) [32,33] was used to identify and characterize the virulence genes within the genome. The antibiotic resistance genes were identified using the Comprehensive Antibiotic Resistance Database (CARD) [34]. The mobile genetic elements in the DNA sequences of the strain were identified through the web-based tool oriTfinder (https://tool-mml.sjtu.edu.cn/oriTfinder/oriTfinder.html) (accessed on 2 January 2024). All the above predictions were made using default parameters.

### 2.3. Phenotypic Safety and Probiotic Characteristics Assessment

#### 2.3.1. Acid and Bile Salt Tolerance

To assess acid tolerance, 2% (*v*/*v*) activated suspensions of *E. faecium* B13 were inoculated into MRS medium, cultured overnight, and centrifugated. The cell pellet was washed twice with PBS and resuspended in PBS of pH 1.0, pH 2.0, pH 3.0, pH 4.0, and pH 7.0, respectively, then incubated at 37 °C for 3 h. The viable cells were counted using the flat colony counting method. The number of viable bacteria was assessed on MRS agar, and the results are expressed as log_10_ CFU/mL. The survival rate of the bacteria was calculated using Formula (1):Survival rate (%) = N1/N0 × 100.(1)
where N1 (log CFU/mL) is the total viable cell after treatment (3 h), and N0 (log CFU/mL) represents the total viable cell at pH 7.0.

To assess bile tolerance, 100 µL cell suspensions of B13 cultured overnight were added into MRS broth containing 0.3% (*w*/*v*) bile salt and incubated at 37 °C. The viable bacteria were measured after 0, 1, 2, and 3 h of exposure to bile salt treatments. The survival rate was calculated using Formula (2):Survival rate (%) = Nt/N0 × 100.(2)
where Nt (log CFU/mL) is the total viable cell after different treatments, and N0 (log CFU/mL) represents the total viable cell before treatment (0 h).

#### 2.3.2. Cell Surface Hydrophobicity

The cell surface hydrophobicity (CSH) of the strain was determined by bacterial adhesion to hydrocarbons (BATH) [35]. *E. faecium* B13 in the logarithmic growth phase was harvested by centrifugation (6000 rpm, 4 °C, 15 min), washed twice with PBS (pH 7.0), and resuspended in the same solution. The n-hexadecane and bacterial suspensions were mixed at a ratio of 1:5 (*v*:*v*) and vortexed for 2 min. The mixture was kept at room temperature for 45 min, and then the OD_600_ of the aqueous phase was determined. CSH was calculated using Formula (3):CSH (%) = (A_0_ − A/A_0_) × 100.(3)

A_0_ and A represent the OD_600_ values measured before and after mixing with n-hexadecane.

#### 2.3.3. Auto-Aggregation

The auto-aggregation analysis followed the procedure described by Reubene et al. [36], with some modifications. Overnight cultures were centrifuged (6000 rpm, 5 min) and washed twice with PBS. The sediment was resuspended in PBS and vortexed for 10 s, and then the OD_600_ was measured as A_0_ (initial optical density). The bacterial suspensions were incubated at 37 °C and measured at 0, 1, and 5 h as A_1_. Auto-aggregation ability was calculated as follows (4):Auto-aggregation (%) = [1 − (A_1_/A_0_) ] × 100.(4)

#### 2.3.4. Antioxidant Activity and Total Antioxidant Capacity

The 2,2-diphenyl-1-picrylhydrazyl (DPPH) free-radical scavenging activity of *E. faecium* B13 fermented supernatant was assessed according to the method described by Wu et al. [37] with minor modifications. The culture medium of *E. faecium* B13 (24 h) was centrifugated at 4 °C (6000 rpm for 5 min) and filtered through a 0.22 μm filter membrane to obtain the cell-free supernatants (CFS). Then, 1.5 mL of CFS was mixed with 1.5 mL of ethanolic DPPH solution (0.2 mM). DPPH solution was mixed with sterilized water as a control group, and CFS was mixed with ethanol as a blank group. All groups were incubated in the dark at 25 °C for 30 min, and then the absorbance was measured at a wavelength of 517 nm. The DPPH free-radical scavenging activity of VC (1 mmol/L) was assessed at the same condition as the positive control. The scavenging ability was calculated according to Formula (5):Scavenging activity (%) = [1 − (A_sample_ − A_blank_)/A_control_] × 100.(5)
where A_sample_, A_blank_, and A_control_ represent the absorbance at 517 nm of the sample, blank, and control groups.

The total antioxidant capacity was measured according to the instructions of the total antioxidant capacity (T-AOC) test kit (Nanjing Jiancheng Bioengineering Research Institute, China). The total antioxidant capacity of VC was measured as a positive control.

#### 2.3.5. Bile Salt Hydrolase (BSH) Activity

The bacteria were streaked on modified MRS agar and added extra 0.5% (*w*/*v*) sodium salt of Tauro deoxycholic acid (TDCA) (Sigma-Aldrich, St. Louis, MO, USA) and 0.37 g/L of calcium chloride (Sigma-Aldrich, USA). Plates were incubated at 37 °C for 72 h. A halo zone around colonies was observed (indicating bile salt hydrolase activity existed).

#### 2.3.6. Gelatinase Activity

The bacterial suspensions were added into the gelatin biochemical identification tube (Qingdao Hope Bio-Technology Co., Ltd., Qingdao, China) and cultured at 37 °C for 48 h. Then the tube was placed at 4 °C for 30 min, and solidification was observed by slanting. Whether liquefaction occurred in the medium was the standard for determining gelatinase activity. If the medium is liquid, the gelatinase activity is positive. *S. aureus* ATCC 25923 in a gelatin tube and PBS in a gelatin tube were used as positive and negative controls.

#### 2.3.7. Decarboxylase Activity

The *E. faecium* B13 was respectively inoculated into the double-arginine hydrolase broth, the ornithine decarboxylase broth, the lysine decarboxylase broth, and the corresponding control broth (Qingdao Hope Bio-Technology Co., Ltd., Qingdao, China), then cultured at 37 °C for 24 h. *E. coli* 8099 and MRS broth were used as positive and negative controls. If the color of the test medium and the control medium did not change, decarboxylase activity is negative. If the color of the test medium changed to purple and the color of the control medium was still yellow, decarboxylase activity is positive.

#### 2.3.8. Indole Experiment

The *E. faecium* B13 was cultured at 37 °C for 24 h in tryptophan broth. Then Kovac’s reagent was added and shaken. If the color of the medium changes to rose, the tryptophanase activity is positive. *E. coli* 8099 and MRS were used as positive and negative controls.

#### 2.3.9. Hemolysin Activity

The *E. faecium* B13 was streaked on the columbia blood agar plate and cultured for 48 h [38]. Then, a hemolytic transparent circle around the colony was observed. The hemolytic *S. aureus* ATCC 25923 was used as the positive control. If grass green ring appeared around the colony, it is α hemolysis; if a clear ring appeared around the colony, it is β hemolysis; if the culture medium around the colony had no change, it is γ hemolysis, which means no hemolysis.

#### 2.3.10. Antibiotic Susceptibility

According to the Kirby Bauer (K-B) method recommended by the American Clinical Laboratory Standardization Institute (CLSI), the diameter of the inhibition zone of 17 kinds of antibiotics against *E. faecium* B13 was measured to determine the drug resistance. *S. aureus* ATCC 25923 was used as a quality control strain in this test [39].

#### 2.3.11. Cell Adhesion In Vitro

Caco-2 cells were cultured at 37 °C in a humidified environment containing 5% CO_2_ in DMEM medium containing 10% fetal bovine serum (Gibco, Shanghai, China) and seeded in 24-well tissue culture plates and incubated until full differentiation (21 days).

Bacterial suspensions at a concentration of 10^8^ CFU/mL were applied to confluent Caco-2 monolayers [40]. After 1 h of incubation at 37 °C, the mixture was rinsed three times with PBS, and 0.5% Triton X-100 solution (Sigma-Aldrich, Shanghai, China) was added to digest the cells. The digested cell suspensions were placed on MRS agar to determine the number of adherent bacteria. The adhesion percentages were calculated by comparing the viable cell numbers before and after adhesion.

#### 2.3.12. Inhibitive Pathogen Adhesion

Three different methods were used to examine the ability of *E. faecium* B13 to defend against pathogen adhesion to Caco-2 cells [41]. The competition was assessed in group 1. 1 × 10^8^  CFU/mL of *E. faecium* B13 and 1 × 10^8^ CFU/mL of *E. coli* were added simultaneously to the CaCo-2 cells and incubated at 37 °C for 2 h. The displacement was assessed in group 2. CaCo-2 cells were incubated with *E. coli* suspensions for 1 h and then added to *E. faecium* B13 suspensions and incubated together for 1 h. The exclusion was assessed in group 3. CaCo-2 cells were incubated to *E. faecium* B13 suspensions for 1 h, then added with *E. coli* suspensions for 1 h. CaCo-2 cells were incubated with  1 × 10^8^  CFU/mL of *E. coli* for 2 h as the control group. All groups were rinsed with PBS (pH 7.2) and treated with a 0.5% Triton X-100 solution (Sigma-Aldrich, Shanghai, China). The digested cell suspensions were serially diluted in PBS, and colonies were counted on MacConkey Agar. The inhibition rate was calculated using Formula (6):Inhibition rate (%) = (1 − (N_t_/Nc)) × 100.(6)
where N_t_ (log CFU/mL) represents viable cells in the treated group and Nc (log CFU/mL) represents viable cells in the control group.

### 2.4. Effects In Vivo

#### 2.4.1. Animals and Bacteria

Male ICR mice (3 weeks) were housed under standard conditions with an alternating 12 h light and dark cycle at a temperature of 25  ±  2 °C and with free access to food and water. *E. faecium* B13 was cultured at 37 °C for 24 h in the MRS medium, then centrifuged for 10 min at 6000 rpm, rinsed, and resuspended in normal saline.

#### 2.4.2. Intestinal Distribution of E. faecium B13

To investigate the characteristics of *E. faecium* B13 colonization of the GIT, fluorescein isothiocyanate (FITC) was used to stain B13 [42]. The *E. faecium* B13 was collected during the logarithmic growth phase, washed three times with normal saline, and resuspended in the same solution. An equal volume of FITC staining solution (Beyotime, Shanghai, China) was added to the bacterial suspensions, and the mixture was incubated for 30 min at 37 °C in the dark. The samples were then rinsed three times to remove the unbound dye. B13 was resuspended in normal saline at a concentration of 1 × 10^10^  CFU/mL.

The entire experimental period was 14 days. The mice were gavaged with 200 μL of fluorescent-labeled bacterial suspensions for 7 consecutive days, then stopped on the 8th day, whereas the control group was given normal saline the whole time. Mice were euthanized at 2 h, 4 h, 6 h, 8 h, 12 h, 24 h, and every day to collect 1 cm sections of the jejunum, ileum, colon, and cecum. After thoroughly rinsing the intestinal tube with normal saline, the intestine homogenates were centrifuged, and the precipitate was resuspended to detect the fluorescence value. Meanwhile, the sections of ileum and colon on the 4th day were fixed in formalin, processed in paraffin, and stained with DAPI.

#### 2.4.3. Animals Experiment

The safety experiment of *E. faecium* B13 in vivo was conducted for 28 days. After a one-week adaptation period, mice were randomly divided into 2 groups of 8 animals each. Control group: the mice were gavaged with 200 μL of normal saline once a day. Experimental group: the mice were gavage equivalent bacterial suspensions (5 × 10^9^ CFU/mL).

The mice were monitored daily for changes in clinical signs, mortality, fur, body weight, and food intake. On the 29th day, the mice were fully anesthetized for sacrifice after a fast of 12 h. The blood samples were being taken for routine examination. The organs, such as the heart, liver, heart, spleen, lung, and kidney, were weighed to calculate the organ index and determine the viable microbe count on these organs. Colon was collected for histological analysis by hematoxylin and eosin (H&E) staining. Colon contents were collected for further analysis. All the procedures involving animals followed the guidelines of the national standards outlined in GB 14925–2010 [43] and permitted by the Southwest Minzu University Animal Ethics Committee.

#### 2.4.4. Metagenome Sequencing of Colon Microbiota

Colon contents (*n* = 5) were collected and sequenced. The metagenomic sequencing was based on Shanghai Majorbio Bio-Pharm Technology Co. Ltd. and was in accordance with the company’s standard protocols (Shanghai, China).

### 2.5. Statistical Analysis

All measurements were repeated independently in triplicate, and the results are expressed as the mean ± standard deviation. Statistical analyses were performed using SPSS version 25.0. The data were subjected to a one-way analysis of variance (ANOVA), followed by a Duncan’s test or Student’s *t*-test to examine for significant differences. Differences in bacterial microbiota community structures between samples were visualized by principal coordinate analysis (PCoA). The difference was considered significant at *p* < 0.05.

## 3. Results and Discussion

### 3.1. The Whole Genome Sequence Analysis of the E. faecium B13

#### 3.1.1. General Genome Features

A total of 14,188,932 raw reads of the *E. faecium* B13 strain were generated by Illumina Novaseq 6000 mode. The Q30 value of raw reads was 91.29%. After data QC and filtering, approximately 2040 Mb of clean data were used for assembly. The *E. faecium* B13 genome, with an average GC content of 38.35%, consists of one chromosome and two plasmids (plasmid1 and plasmid2). The chromosome contains 2,591,601 bp, while plasmid1 and plasmid2 contain 197,836 bp and 37,755 bp, respectively (Figure 1). The whole genome consisted of 2785 genes, including 68 tRNAs, 6 5S rRNAs, 6 16S rRNAs, 6 23S rRNAs, and 52 other ncRNAs.

#### 3.1.2. Functional Annotation

A total of 2390 protein-coding genes (85.81% of the total protein-coding genes) were assigned a putative function by COG, which belong to 20 different categories, respectively. These gene categories involved replication, transcription, translation, transport, and metabolism of carbohydrates, nucleic acids, and lipids, as well as material transport and energy conversion. Functional genes associated with transcription (254 Open Reading Frames (ORFs)), translation (206 ORFs), and carbohydrate transport and metabolism (246 ORFs) were ranked among the most abundant COG functional categories. The functional categories of genes annotated are shown in Figure 1. The research by Michael S. Gilmore [44] showed that the largest functional gene groups that helped *E. faecalis* adapt to the environment were those genes involved in translation, ribosome structure, and biogenesis. The diversity of functional annotations indicated that their genomes have high plasticity to adapt to different environments.

#### 3.1.3. Average Nucleotide Identity

According to ANI analysis (strain information is shown in Appendix A), the closest genetic relationship to *E. faecium* B13 is the strain DUTYH_16120012 isolated from Chinese sauerkraut (ANI = 99.44); in addition, the strains isolated from Korean fermented soybean also have a high ANI value near 99% (98.87%~99.14%) with *E. faecium* B13. However, compared to those strains selected from other foods, such as yogurt, cheese, fermented yak milk, fermented milk, camel milk, and commercial food with *E. faecium* B13, the ANI value was relatively lower, about 95% (94.68%~95.28%). Moreover, the similarity between *E. faecium* B13 and other natural environment strains, such as clinical isolation, wastewater, and soil, was also low (94.71%~95.34%). Surprisingly, *E. faecium* B13 was very similar (99.33%) to a probiotic candidate 17OM39 isolated from the feces of a healthy adult in India [45].

These results indicated that the similar living environment and fermentation substrate may be the cause of the high similarity between Chinese sauerkraut, Korean fermented soybean, and *E. faecium* B13. Many studies have shown that *E. faecium* derived from fermented foods has higher safety [4,46] than clinical strains. Genomes of clinical strains are significantly larger than those of non-clinical strains because of the acquisition of mobile genetic elements, virulence, and AMR genes [4,47]. Exceptionally, further analysis showed that the potential virulence genes and drug resistance genes carried by B13 were significantly different from 17OM39, which may be caused by the two circular plasmids carried by *E. faecium* B13.

#### 3.1.4. Virulence Genes

The results of virulence genes annotated by the virulence factors of the database (VFDB) are shown in Table 1. A total of 8 genes related to virulence were specifically annotated, involving these functions: biofilm formation, adhesion, streptolysin, and evasion of the immune system. Among them, 5 genes were located on the chromosome, and the rest were located on plasmid 1. The strain carried multiple pili genes (*pilA*, *pilB*, *pilE*, and *pilF*). Although pili genes are associated with the pathogenicity of certain bacteria, they can also help bacteria adhere to host cells. Adherence to the gut epithelial cells and subsequent colonization could extend the persistence of probiotic strains in the intestinal tract. The *bopD* encoded the secretory protein of the type III secretion system, which is a transmembrane channel formed by a multi-component protein complex and is a complex molecular device presented in many Gram-negative pathogenic bacteria [48]. The *bopD* locus is regulated by the Fsr system and is necessary for biofilm formation [49,50]. The strain does not carry major virulence genes that the clinical *E. faecium* strains possess, like *asa1, gelE, cylA, esp*, and *hyl*, suggesting that the strain is unlikely to initiate opportunistic infection.

Compared to these *E. faecium* strains isolated from commercial swine and cattle probiotic products [47], B13 contains fewer virulence genes. But the *sagA* gene was identified to suggest the strain may secrete SLS, which can cause β-hemolysis. So, the safety of B13 needs further testing.

#### 3.1.5. Antibiotic Resistance Genes

The antibiotic-related genes of *E. faecium* B13 were detected using CARD databases and shown in Table 2. *E. faecium* B13 carries genes encoding resistance to a few medically important antibiotics, including aminoglycosides (*aac(6′)-Ii*, *ant(6)-Ia*), macrolide (*msrC*), lincosamide (*lnu(G)*), pleuromutilin (*eatA*), and the tetracyclines *tet(L)* and *tet(M)*. Genetic traceability analysis traced that *tet(L)* and *tet(M)* genes may originate from *Geobacillus stearothermophilus* and *Staphylococcus aureus*, respectively; *ant (6)-Ia* genes may originate from *Exiguobacterium*.

Tetracycline resistance is commonly present in Gram-positive and Gram-negative bacteria and is easily transmitted in the environment. Research by Sunghyun Yoon [1] showed that up to 73.4% of 338 strains of enterococci isolated from 1584 batches of bulk tank milk samples were resistant to tetracycline from 396 farms that were affiliated with four dairy companies in Korea. Genomic analysis of antibiotic-resistant *Enterococcus* spp. revealed the spread of plasmid-borne *tet(M)*, *tet*(L), and *erm(B)* genes from chicken litter to agricultural soil in South Africa by Fatoba [51]. Even foodborne lactobacillus was able to spread *tet(M)*, *tet(L)*, and *tet(W)* in different natural environments [52].

The *lnu (G)* gene located on a mobile element (Tn 6260) is easily disseminated [53]. In our study, these genes (*tet(L)*, *tet(M)*, *ant (6)-Ia,* and *InuG)* are all located in plasmids and may get rid of by plasmid elimination.

Unlike *E. faecalis, E. faecium* is naturally susceptible to lincosamides, streptogramins A, and pleuromutilins (LSAP phenotype). But the *eat (A)* gene is an intrinsic gene of *E. faecium,* and the *eat (A)* protein displayed 66%, 44%, 43%, and 42% amino acid identities with other proteins *Lsa (A)*, *Lsa (E)*, *Lsa (B)*, and *Lsa (C),* conferring LSAP-type resistance in various Gram-positive organisms. So, when this gene undergoes a mutation, the strain may exhibit the LSAP phenotype.

The bacteriocin gene has been predicted through WGS analysis and had also been detected [54], so the liaFSR system and *cls* gene may definitely be inherent systems. Because *E. faecium* is Gram-positive, it is intrinsically resistant to low levels of aminoglycoside [55]. Aminoglycoside phosphotransferase (APH), aminoglycoside riboside transferase (ANT), and aminoglycoside acetyltransferase (AAC) can modify the amino groups or hydroxyl groups of aminoglycoside antibiotics and destroy their binding to the ribosome, thus rendering aminoglycoside antibiotics unable to work [56].

The presence of *aac(6′)-Ii* and *ant (6)-Ia* genes suggests a moderate and high level of aminoglycoside resistance. The strain possessed the *msrC* gene, which allows the organisms to be resistant to macrolides, but when *aac (6′)-Ii* and *msrC* coexist, they may be non-functional, which was present in many macrolide-susceptible strains [57].

### 3.2. Probiotic Properties Assessment

#### 3.2.1. Acid and Bile Salt Tolerance

Tolerances to low-acid and high-bile salt stresses are significant properties for any potential probiotic bacteria. The abilities of *E. faecium* B13 to tolerate acid and bile salts are presented in Table 3 and Table 4. The pH of animal gastric juice can fluctuate from around 1.0 on an empty stomach to 4.0 after eating and even reach 5.0 after consuming yogurt and fermented milk. The survival rates of B13 at different pH conditions varied greatly (*p* < 0.05) from 0 at pH 1.0 and 2.0 to above 97.97% at pH 3.0 to pH 4.0. Its acid tolerance was similar to that of *E. faecium* MG5232 from Kimchi [5], but far below that of MZF1-MZF5 selected from artisanal Tunisian meat [40] and *E. faecium* OV3-6 from the fermentation plant [41]. The results showed that the bile salt tolerance of B13 significantly decreased with time (*p* < 0.05), but the survival rates at different times were all found to be above 90%. The bile salt tolerance of B13 was similar to 17OM39 from human feces [45]. In addition, B13 exhibited significantly high survival rates in 0.3% bile salt solution compared to some *Pediococcus* species [58,59,60,61]. WGS analysis detected the *bsh* gene, and the bile salt hydrolase activity was positive. The high tolerance to bile salt of B13 may be due to its bile salt hydrolase. *E. faecium* OV3-6 showed BSH activity and resistance to simulated small intestine conditions, with the percentage survival of the strain above 96.89% at 4 h. The good tolerance for acid and bile salt suggested that B13 could potentially reach the intestinal lumen and thus stay alive in that environment.

#### 3.2.2. Antioxidant Activity In Vitro

A lot of research has shown that the fermented supernatant of the strain had significantly higher DPPH radical scavenging activity compared to its intact cells [37]. So, our study assessed the antioxidant capacity of the fermentation supernatant. The free radical DPPH clearance rate and the total antioxidant capacity of *E. faecium* B13 (Table 5), respectively, were 32.83% and (19.28 ± 3.14) U/mL, lower than those of the vitamin C group (88.39% and 37.45 ± 1.35 U/mL). Numerous lactobacillus, such as *Lactiplanti bacillus plantarum* GXL94 [62] and LGG [63], have been reported to exhibit high DPPH scavenging activity. Unlike these *lactobacilli*, the antioxidant capacity of *E. faecium* has rarely been reported. The antioxidant activities could increase in a dose-dependent manner [64]. Therefore, increasing the bacterial concentration may enhance the antioxidant capacity of the fermented supernatant of B13.

#### 3.2.3. Cell Adhesion

The adhesion rate of B13 to Caco-2 cells was 36.69% at 37 °C. Compared with *E. faecium* GEFA01 (above 30%), *E. faecium* 17OM39 (57%), *E. faecium* MZF5 (21%), and *E. durans* WEDU02 (7.50%), the bacterial adhesion rate to Caco-2 is related to the hydrophobicity and auto-aggregation of bacteria, unrelated to the bacterial source. Interestingly, Mohamed Zommiti et al. [40] reported that auto-aggregation ability had no direct correlation with the adhesion rate. Moreover, the inhibition of *E. coli* adhesion to Caco-2 cells by *E. faecium* B13 is shown in Figure 2B. The results indicated that the adhesion sites of the two bacteria had a certain degree of duplication.

#### 3.2.4. Hydrophobicity and Auto-Aggregation

The hydrophobicity and auto-aggregation of bacteria are indicators of the non-specific adhesion of probiotics to the intestine. Auto-aggregation and hydrophobicity help bacterial adhesion to epithelial cells of the host GIT and the prevention of pathogen colonization [65] and also help biofilm formation by LAB, which further promotes colonization [66].

B13 showed a low level of hydrophobicity (11.3%). The hydrophobicity of B13 is equivalent to that of *E. faecium* MG5232 from Kimchi, which is lower than that of 17OM39 from human feces and *Enterococcus durans* SJRP29 from cheese [3]. Compared to previous studies, the autoaggregation ability of various probiotic strains was approximately 30–96%, with an average of 62.6% [67]. Our result showed that the auto-aggregation ability increased with time, changing from 20% at 2 h to 77% at 24 h (Figure 2A). The same result was reported by other research [68]. In addition, the auto-aggregation was proven to be strain specific and may vary in the same taxonomic group [69,70].

### 3.3. Safety Assessment In Vitro

The *E. faecium* B13 did not exhibit gelatinase (Figure 2C), ornithine decarboxylase (Figure 2D), or tryptophanase activity, which are consistent with genotype results. Variously, WGS detected the *sagA* gene and *LOG* gene; there was no *LOC* gene in B13, but *E. faecium* B13 had no hemolysis (Figure 2E), exhibited arginine decarboxylase activity. These phenotypes cannot match genotypes. Arginine decarboxylase can catalyze arginine to spermine, which has a significant physiological regulatory effect at low concentrations [71]. Some edible microorganisms and probiotic strains were reported to produce biogenic amines [72,73,74], which have a significant impact on the flavor of food.

The antibiotic sensitivity test showed that B13 was sensitive to 12 antibiotics but resistant to 5 antibiotics, including erythromycin, rifampicin, tetracycline, doxycycline, and minocycline (Table 6).

The results of the antibiotic sensitivity test were consistent with the predicted results of antibiotic resistance genes, except for two cases. In our study, B13 carried the gentamicin resistance genes (*tet (L)* and *tet (M)* (belonging to aminoglycosides) detected by whole genome sequencing, but it is phenotypically susceptible. Genetic modification or silence may cause the discrepancies between gene predictions and phenotypic experiments [75]. Some studies have identified individual strains with silenced AR (antibiotic resistance) genes [76]. But dormant AR genes are able to resume their expression in certain environments [77]. Furthermore, this strain was phenotypically resistant to rifampicin but did not possess related AR genes. It is possible that the strain had nonspecific efflux pumps or had other new resistance genes that had not been discovered. Additionally, gene prediction has some limitations. Its accuracy is affected by many complex factors, such as detection depth, database breadth, and transcriptional changes [78].

These results suggested genotype–phenotype discrepancies were relatively common in *E faecium*. The study on 197 strains of *Enterococcus* [57] showed genotype-to-phenotype correlations of 97% for *E. faecalis* but only 88% for *E. faecium*.

### 3.4. Intestinal Localization of E. faecium B13

Fluorescent-labeled bacteria could be observed in all intestinal segments after 2 h of gavage. The fluorescence value reached a maximum at 6 h in the colon and cecum and at 2 h in the jejunum and ileum, then declined within 24 h. For one week of feeding, the fluorescence value continued to increase and reached its maximum on the 3rd day of gavage, maintained until the second day after stopping gavage. (Figure 3A,B). The localization of B13 in the intestine is shown in Figure 3C.

### 3.5. Animal Experiment

During the experimental period, there were no deaths or other pathological reactions observed in the mice. There was no significant difference in routine blood and organ index between the two groups (Appendix A), and no live bacteria were detected in the heart, liver, spleen, lungs, kidneys. In addition, pathological sections of the mouse colon showed no abnormalities by H&E staining (Figure 3D), further supporting its safety.

### 3.6. Metagenome Sequencing

There was no statistically significant difference among the groups in the α-diversity indices (Appendix A), indicating that *E. faecium* B13 did not change colon microbiota diversity. The structure of the microbiome was analyzed to clarify the role of B13 in the gut microbiome. The gut microbiota of the B13 group mainly consisted of *Firmicutes* (34.30%), *Bacteroidetes* (58.0%), *Proteobacteria* (3.37%), and *Actinobacteria* (2.45%). At the phylum level (Figure 4A), the ratio of *Firmicutes/Bacteroidetes* (F/B) decreased compared with the control group. The differences in microbial composition at the genus level are shown in Figure 4B. The relative abundance of unclassified *Lachnospiraceae* [79], *Candidate Amulumruptor* [80,81], and *Lactobacillus* increased; however, *Prevotella* decreased, but the difference is not significant. Notably, the relative abundance of the species *Heminiphilus faecis* between groups showed a trend of difference (*p* = 0.09) (Figure 4C). *H. faecis* is closely related to the genus *Muribaculum* [82] and belongs to the family *Muribaculaceae*, which is dominant in the mouse gut and beneficial to health [83,84]. *M. intestinale* is a newly cultured species and a potential species associated with a healthy diet and exercise [85,86]. Principal Co-ordinates Analysis (PCoA) revealed no significant difference between the two groups, but the control group showed a dispersion trend, while the B13 group exhibited a more even gut microbial structure (Figure 4D). The impact of microbes on the gut microbiota was strain specific. Some probiotics may regulate microbial community structure [87], some may have an impact on the host even though they do not necessarily interact with indigenous microbiota [88], and some may normalize the disturbed microbiota and modulate it beneficially [89,90]. For example, the influential mechanism of *lactobacillus rhamnoses* M9 [91] in the intestines is different from that of *Lactobacillus fermentum* HNU312 [92]. Furthermore, microbial influence also exhibits a time effect, such as the different effect of *R. intestinalis* on gut microbiota on 7 days and 14 days [68]. 

In a word, B13 cannot significantly regulate gut microbiota during 28 days, but it can stabilize the core microbial community and has a beneficially regulatory effect on gut ecology.

As B13 contains plasmids, there is a risk of drug resistance gene transfer. Therefore, the ARGs were analyzed between the two groups. A total of 532 ARGs were detected in the *E. faecium* B13 group and 538 ARGs in the control group. ARGs with an abundance of over 1% showed no significant difference between groups, and the drug resistance genes carried by B13 did not increase. In addition, the oriT was not identified in the genome. These results proved from phenotype and genotype levels that B13 did not transmit resistance genes during the feeding cycle.

## 4. Conclusions

Since *Enterococcus* spp. has been considered a category of opportunistic pathogens, *E. faecium* B13 needs a more comprehensive safety evaluation before use. In this paper, WGS revealed the genetic background of B13 and demonstrated its safety. The strain performed excellent probiotic properties in vitro, such as high tolerance to acid and bile salt, high antioxidant activity, etc. Non-hemolytic activity, non-gelatinase activity, and limited antibiotic resistance confirmed its risk was under control. Study in vivo and metagenome sequencing further proved the probiotics and safety of B13. So, *E. faecium* B13 shows promise as a probiotic, and further metabolism analysis in the intestines will be conducted soon.

## Figures and Tables

**Figure 1 microorganisms-12-00994-f001:**
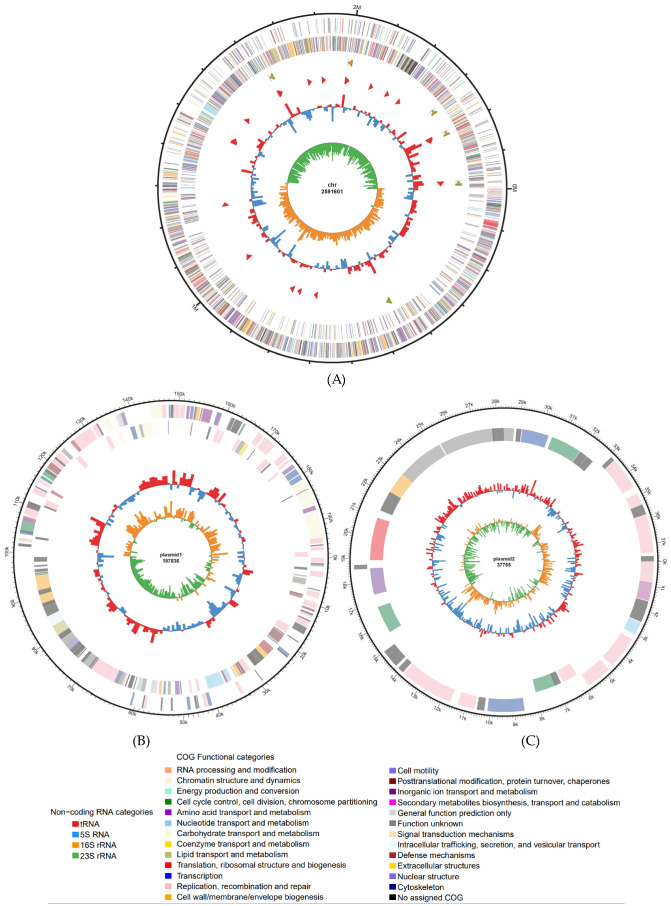
The complete genome of *E. faecium* B13. (**A**) Chromosomal genome map of *E. faecium* B13. (**B**) Plasmid 1 genome map of *E. faecium* B13. (**C**) Plasmid 2 genome map of *E. faecium* B13. The outermost circle of the circle map is the genome-sized logo; each scale is 0.1 Mb (for Figure 1A), 10 kb (for Figure 1B), and 1 kb (for Figure 1C). The second and third circles are CDS on the forward and reverse chains, and the assorted colors indicate different COG classifications of the CDS. The fourth circle is rRNA, or tRNA. The fifth circle is the GC content, and the outward orange part indicates that the GC content in the region is higher than the whole-genome average GC content. The inward green portion indicates that the GC content in the region is low; the higher peak value indicates a greater difference from the average GC content.

**Figure 2 microorganisms-12-00994-f002:**
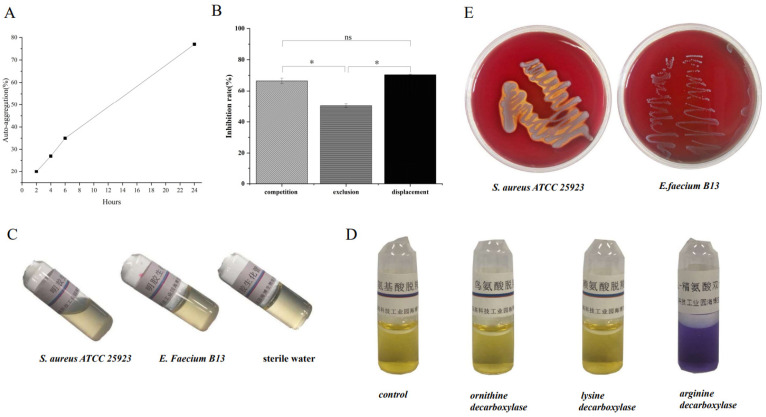
Probiotic properties and safety assessment in vitro. (**A**) Auto-aggregation abilities of *E. faecium* B13. (**B**) Inhibition of *Escherichia coli* adhesion to Caco-2 cells by different modes. (**C**) Gelatinase activity of *E. faecium* B13 compared to *S. aureus* ATCC25923 (positive control) and sterile water (negative control). (**D**) Decarboxylase activity. (**E**) Hemolytic activities. * Indicates *p* < 0.05; ns indicates no significance.

**Figure 3 microorganisms-12-00994-f003:**
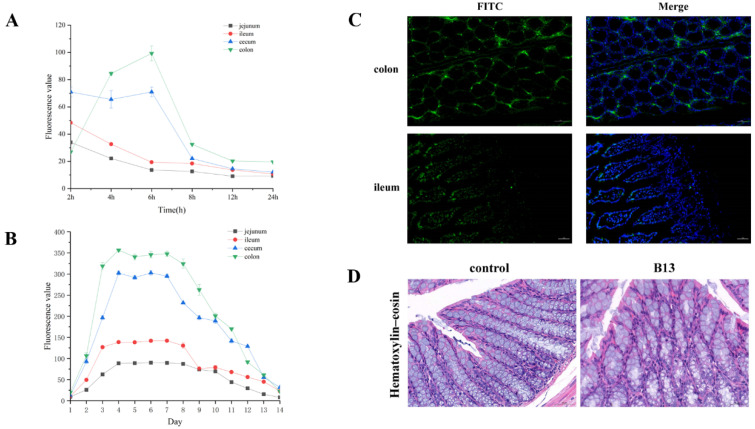
Fluorescence changes and histological sections of *E. faecium* B13 in the intestine. (**A**) Fluorescence values in intestines during 24 h. (**B**) Fluorescence values in the intestines during the whole experiment. (**C**) Paraffin sections were used to observe the distribution of *E. faecium* B13 in the colon and ileum. (**D**) Hematoxylin–eosin staining of the colon. scale bar: 50 μm.

**Figure 4 microorganisms-12-00994-f004:**
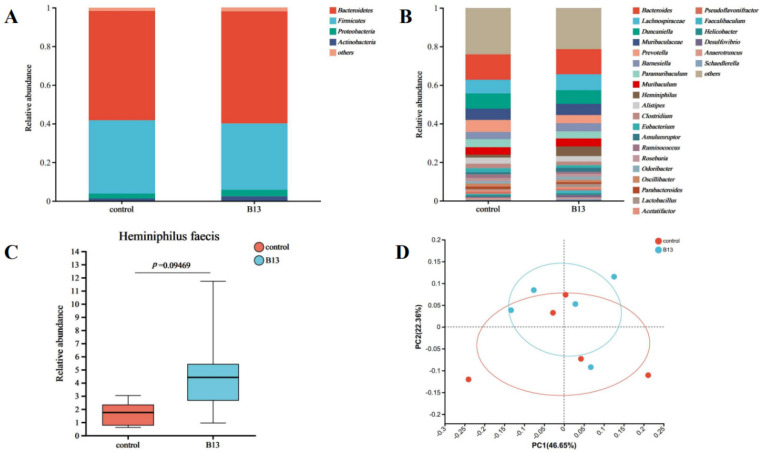
Microbial community composition in the intestine. (**A**) Classification of microflora at the phylum level. (**B**) Classification of microflora at the gene level. (**C**) The relative abundance of *Heminiphilus faecis* between B13 and control groups. (**D**) PCoA analysis between the control and B13 groups.

**Table 1 microorganisms-12-00994-t001:** Virulence gene of *E. faecium* B13.

Virulence genes	Product	Function	Location	Identify(%)	VFid	GeneID
Adherence						
pilB	pilB-type pili	Involved in type IV pili biosynthesis	chr	99.2	VFG042991	B13000560
pilF	minor pilin subunit	Involved in type IV pili biosynthesis	plasmid1	99.1	VFG042984	B13002625
pilE	cell wall-associated LPXTG-like protein	Involved in type IV pili biosynthesis	plasmid1	99.2	VFG042986	B13002627
pilA	pilA-type pili	Involved in type IV pili biosynthesis	plasmid1	98.3	VFG042988	B13002629
Biofilm formation						
sgrA	cell wall anchored protein SgrA	Adherence to cell wall	chr	79.6	VFG043511	B13001199
bopD	Sugar binding transcriptional regulator	biofilm on plastic surfaces	chr	86.3	VFG002197	B13000384
Immune Evasion						
hasC	Hyaluronic acid HA capsule	Evasion of host immune system	chr	73.3	VFG005865	B13001994
pathogenicity						
sagA	Streptolysin S core peptide	hemolytic activity	chr	95.4	VFG043441	B13002370

**Table 2 microorganisms-12-00994-t002:** Resistance genes of the *E. faecium* B13 genome by CARD analysis.

Location	Identify(%)	Gene family	Product	Drug Class	Resistance Mechanism	Origin Species
plasmid2	99.75	tet(L)	tetracycline efflux MFS transporter Tet(L)	tetracycline	antibiotic efflux	*Geobacillus stearothermophilus*
chr	98.901	AAC(6′)-Ii	aminoglycoside N-acetyltransferase AAC(6′)-Ii	aminoglycoside	antibiotic inactivation	*Enterococcus faecium*
plasmid1	99.625	lnu(G)	lincosamide nucleotidyltransferase Lnu(G)	lincosamide	antibiotic inactivation	*Enterococcus faecalis*
plasmid1	100	ANT(6)-Ia	aminoglycoside nucleotidyltransferase ANT(6)-Ia	aminoglycoside	antibiotic inactivation	*Exiguobacterium*
chr	97.531	liaF	three-component signaling pathway regulator LiaF	lipopeptide	antibiotic target alteration	*Enterococcus faecium*
chr	98.941	cls	cardiolipin synthase Cls	lipopeptide	antibiotic target alteration	*Enterococcus faecium*
chr	99.524	liaR	response regulator transcription factor LiaR	lipopeptide	antibiotic target alteration	*Enterococcus faecium*
chr	100	liaS	sensor histidine kinase LiaS	lipopeptide	antibiotic target alteration	*Enterococcus faecium*
plasmid2	95.149	tet(M)	tetracycline resistance ribosomal protection protein Tet(M)	tetracycline	antibiotic target protection	*Staphylococcus aureus*
chr	97.154	msrC	ABC-F type ribosomal protection protein Msr	macrolide/streptogramin	antibiotic target protection	*Enterococcus faecium*
chr	99	eatA	ABC-F type ribosomal protection protein Eat(A)	pleuromutilin	antibiotic target protection	*Enterococcus faecium*

**Table 3 microorganisms-12-00994-t003:** Acid tolerance of *E. faecium* B13 at different pHs.

pH	Viable Count (log10 CFU/mL)	Survival Rate (%)
1.0	-	0
2.0	-	0
3.0	6.74 ± 0.03	97.97
4.0	6.95 ± 0.04	101

“-” represents the number of viable bacteria is 0.

**Table 4 microorganisms-12-00994-t004:** Resistance of *E. faecium* B13 at 0.3% bile salt conditions.

Times of Exposure (h)	Viable Count (log10 CFU/mL)	Survival Rate (%)
1	7.80	95%
2	7.61	92.69%
3	7.45	90.62%

**Table 5 microorganisms-12-00994-t005:** Antioxidant activity of *E. faecium* B13 and Vc.

Subjects	CFS	Vitamin C
DPPH radical (%)	32.83 ± 0.37	88.39 ± 1.64
total antioxidant capacity (U/mL)	19.28 ± 3.14	37.45 ± 1.35

**Table 6 microorganisms-12-00994-t006:** Antibiotic susceptibility of *E. faecium* B13.

Antibiotic	ZOI (mm)	Antibiotic Susceptibility	Antibiotic	ZOI (mm)	Antibiotic Susceptibility
Nitrofurantoin	21.76 ± 0.93	S	Teikoplanin	22.4 ± 0.6	S
Zyvox	28.9 ± 1.3	S	Gentamicin	20.8 ± 0.4	S
Erythromycin	12.0 ± 0.6	R	Doxycycline	8.0 ± 0.1	R
Vancomycin	25.8 ± 0.5	S	Minocycline	8.9 ± 0.1	R
Rifampicin	11.1 ± 0.6	R	Gatifloxacin	25.1 ± 0.5	S
Tetracycline	5.2 ± 0.1	R	Norfloxacin	22.0 ± 0.1	S
Ampicillin	25.3 ± 0.6	S	Penicillin	17.7 ± 1.4	S
Ciprofloxacin	24.8 ± 0.4	S	Levofloxacin	21.6 ± 0.4	S
Chloramphenicol	25.9 ± 1.8	S			

S—susceptible; I—intermediate; R—resistant; ZOI—zone of growth inhibition.

## Data Availability

The raw sequences generated for this study can be found in the NCBI Short Read Archive under Bio Project PRJNA1029823 (WGS) and PRJNA1039608 (metagenome).

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
