# Peer review of "Assessment of the Safety and Probiotic Properties of Enterococcus faecium B13 Isolated from Fermented Chili"

_microorganisms, 2024, doi:10.3390/microorganisms12050994_

Round 1

Reviewer 1 Report

Comments and Suggestions for Authors

The paper presented by the authors under the title "Assessment of the safety and probiotic properties of Enterococcus faecium B13 isolated from fermented chili" should be considered as a good scientific work providing valuable information on the probiotic properties in vitro of Enterococcus faecium B13, as well as in vivo study and metagenome sequencing provide evidence for the probiotics properties of strain B13. The methodology used in this study is certainly adequate and the results can be considered of significance from the academic and scientific point of view.  

Author Response

Dear Reviewer:

We are honored to receive your recognition for this paper. Special thanks to you for your good comments.

We appreciate for your warm work earnestly, and hope that the correction will meet with approval.

Thank you very much for your comments and suggestions.

Reviewer 2 Report

Comments and Suggestions for Authors

The article is at a high level both professionally and in terms of content, but there are a large number of minor errors (of a formatting nature) that should be removed. This is, for example, frequent omission of spaces - page 2 ....Sichuan, China).It was..., pH7.0respectively, page 3, B13fermented, 0.5%Triton X-100solution, AMR genes[4,46]Exceptionally, furthermore, the frequent absence of gaps 37°C - correctly 37 °C. Furthermore, failure to mention the chemical on page 2 - C6H14N2O7, others mentioned, writing the unit mL - CFU/ml vs CFU/mL and many other situations also in tables 3 and 4, further .... filtered through 0.22 um filter membrane..... 0.22 µm, in table 5 there is confusion in the units - U/mL vs %, 3.6. metagenome - 3.6.  Metagenome.

Comments on the Quality of English Language

English is fine, only minor mistakes need to be corrected.

Author Response

Dear Reviewer:
We are very sorry for our incorrect writing. Based on your comments and suggestions, we have corrected item by item:

Your comments and suggestions: â‘ . Frequent omission of spaces and absence of gaps. â‘¡. Failure to mention the chemical on page 2 - C6H14N2O7, others mentioned. â‘¢. Writing the unit mL - CFU/ml vs CFU/mL and many other situations also in tables 3 and 4, further .... filtered through 0.22 um filter membrane..... 0.22 µm. â‘£. In table 5 there is confusion in the units - U/mL vs %. ⑤. Capitalized, such as 3.6. metagenome – correctly 3.6. Metagenome.

  1. Response to comment â‘ : We have corrected the errors throughout the entire article based on your comment. Please check the original text for specific modifications.
  2. Response to comment â‘¡: We have added its name before its chemical formula. Please check the original text for specific modifications.
  3. Response to comment â‘¢: We have made corrections to the errors throughout the entire article based on your comment. Please check the original text for specific modifications.
  4. Response to comment â‘£: We have revised the confusion you mentioned regarding the units in Table 5. Please check the original text for specific modifications.
  5. Response to comment ⑤: We have corrected the error and proofread and corrected similar errors throughout the entire paper. Please check the original text for specific modifications.

    We appreciate for your warm work earnestly, and hope that the correction will meet with approval.

    Thank you very much for your comments and suggestions.

Reviewer 3 Report

Comments and Suggestions for Authors

Well done:)

Comments on the Quality of English Language

Minor proofreading and editing would be beneficial ex. use of italics.

Author Response

Dear Reviewer:

We are honored to receive your recognition for this paper. Special thanks to you for your good comments. We have carefully edited all the format of the entire paper. Please check the original text for specific modifications.

We appreciate for your warm work earnestly, and hope that the correction will meet with approval.

Thank you very much for your comments and suggestions.
